# Dietary Patterns and Their Sociodemographic and Lifestyle Determinants in Switzerland: Results from the National Nutrition Survey *menuCH*

**DOI:** 10.3390/nu11010062

**Published:** 2018-12-29

**Authors:** Jean-Philippe Krieger, Giulia Pestoni, Sophie Cabaset, Christine Brombach, Janice Sych, Christian Schader, David Faeh, Sabine Rohrmann

**Affiliations:** 1Division of Chronic Disease Epidemiology, Epidemiology, Biostatistics and Prevention Institute, University of Zurich, Hirschengraben 84, 8001 Zurich, Switzerland; jean-philippe.krieger2@uzh.ch (J.-P.K.); giulia.pestoni@uzh.ch (G.P.); sophie.cabaset@uzh.ch (S.C.); david.faeh@uzh.ch (D.F.); 2Institute of Food and Beverage Innovation, Zurich University of Applied Sciences, Campus Reidbach, Einsiedlerstrasse 34, 8820 Waedenswil, Switzerland; broc@zhaw.ch (C.B.); sych@zhaw.ch (J.S.); 3Research Institute of Organic Agriculture (FiBL), Ackerstrasse 113, 5070 Frick, Switzerland; christian.schader@fibl.org; 4Health Division, Nutrition and Dietetics, Bern University of Applied Sciences, Falkenplatz 24, 3012 Bern, Switzerland

**Keywords:** dietary survey, 24-h recall, language region, clustering, multinomial logistic regression

## Abstract

From a public health perspective, determinants of diets are crucial to identify, but they remain unclear in Switzerland. Hence, we sought to define current dietary patterns and their sociodemographic and lifestyle determinants using the national nutrition survey *menuCH* (2014–2015, *n* = 2057). First, we applied multiple factorial analysis and hierarchical clustering on the energy-standardised daily consumption of 17 food categories. Four dietary patterns were identified (“Swiss traditional”: high intakes of dairy products and chocolate, *n* = 744; “Western 1”: soft drinks and meat, *n* = 383; “Western 2”: alcohol, meat and starchy, *n* = 444; and “Prudent”: *n* = 486). Second, we used multinomial logistic regression to examine the determinants of the four dietary patterns: ten sociodemographic or lifestyle factors (sex, age, body mass index, language region, nationality, marital status, income, physical activity, smoking status, and being on a weight-loss diet) were significantly associated with the dietary patterns. Notably, belonging to the French- and Italian-speaking regions of Switzerland increased the odds of following a “Prudent” diet (Odds ratio [95% confidence interval]: 1.92 [1.45–2.53] and 1.68 [0.98–2.90], respectively) compared to the German-speaking regions. Our findings highlight the influence of sociodemographic and lifestyle parameters on diet and the particularities of the language regions of Switzerland. These results provide the basis for public health interventions targeted for population subgroups.

## 1. Introduction

Switzerland is a high-income country characterised by one of the highest life expectancies and one of the lowest prevalence rates of overweight and obesity in the world [1,2]. Significant variations, however, exist between sociodemographic subgroups in the risk of developing chronic diseases, including type 2 diabetes, gastric and liver cancer, coronary heart disease and stroke [2,3]. Mortality rates and causes of death also vary according to age, sex and place of residence [4,5].

Diet is a major modifiable determinant of most chronic diseases, and dietary choices are known to be strongly influenced by sociodemographic and lifestyle determinants [6,7,8]. Therefore, identifying determinants of dietary consumption is critical for testing whether they contribute to previously reported sociodemographic differences in disease prevalence. From a public health perspective, it would enable Swiss authorities to refine nutrition promotion campaign, further elaborate actions for the Swiss Nutrition Strategy 2017–2024 [9], and will assist in health resource allocation.

Until now, sociodemographic and lifestyle differences in food consumption have remained unclear in Switzerland. Mostly, Switzerland lacks national representative data on food consumption and dietary behaviour and has to rely on single-item self-reported nutrition data from the Swiss Health Survey [10,11], or regional studies [12,13]. The recent *menuCH* study provided the first representative data on food consumption and dietary behaviour in Switzerland [14,15,16]. Early results of the study suggested clear sociodemographic differences in food consumption: first, the adherence to the Swiss food-based dietary guidelines differed between men and women, and between the three language regions of Switzerland (German-, French- and Italian-speaking regions) [15]. In addition, the consumptions of certain foods groups, notably beverages, protein-based products and added fats, are different across language regions [15], corroborating previous findings from the Swiss Health Survey [17].

Hence, we sought to define the current dietary patterns in Switzerland and identify their sociodemographic and lifestyle determinants using data from the national nutrition survey *menuCH* [14]. Based on the multiple languages and cultural influences of Switzerland, we expected to identify unique determinants of dietary patterns beyond those usually reported for Western European countries [6,7,8].

## 2. Materials and Methods

### 2.1. Study Design and Setting

The cross-sectional population-based survey *menuCH* was conducted between January 2014 and February 2015 in ten study centres across Switzerland. Swiss residents aged 18 to 75 years old were drawn from a stratified random sample provided by the Federal Statistical Office as described elsewhere [15]. Briefly, 35 strata (7 × 5) covered the seven major areas of Switzerland (Lake Geneva, Midlands, Northwest, Zurich, Eastern, Central, and Southern Switzerland) and five age categories (18–29, 30–39, 40–49, 50–64 and 65–75 years old). From a gross sample of 13,606 individuals, 5496 were successfully contacted by mail or phone, and 2086 accepted to schedule an interview in one of the study centres (38% net participation rate). Main reasons for refusals were lack of time (56%) and lack of interest (28%) [15]. Among the 2086 participants, 2057 had two complete 24-h dietary recalls (24HDRs) and were included in the analysis. A complete flow diagram of the study has been published previously [15].

### 2.2. Ethics

The survey protocol was approved by the lead ethics committee in Lausanne (Protocol 26/13) on 12 February 2013 and by the corresponding regional ethics committees. All procedures followed the guidelines laid down in the Declaration of Helsinki. Written informed consent was obtained from all participants. The survey was registered at the ISRCTN registry under the number 16778734 (https://doi.org/10.1186/ISRCTN16778734).

### 2.3. Dietary Assessment

The first 24HDR was conducted face-to-face and the second one was done two to six weeks later on the phone, as described elsewhere [14]. A team of 15 trained dietitians used the software GloboDiet^®^ (formerly EPIC-Soft^®^, version CH-2016.4.10, International Agency for Research on Cancer (IARC), Lyon, France) [18,19], adapted to Switzerland (GloboDiet^®^ trilingual databases dated on 12 December 2016, IARC, Lyon, France and Federal Food Safety and Veterinary Office, Bern, Switzerland). A book containing 119 series of six graduated portion-size pictures [20] and about 60 common household measures were used to facilitate the quantification of the foods consumed. Compliance of the dietitians to survey-specific standard operating procedures were assessed during the survey [14,15], and data were cleaned and screened for inconsistencies according to IARC’s guidelines [21]. Linkage between foods, recipes and ingredients from GloboDiet^®^ with the most appropriate item from the Swiss Food Composition Database [22] was achieved by the novel matching tool FoodCASE (Premotec GmbH, Winterthur, Switzerland), allowing accessing the macronutrient composition of each food item. As described in Chatelan et al. [15], 81.6% of the study participants were considered as plausible reporters on the basis of their energy intake to basal metabolic rate ratio. Under-reporters (16.9%) and over-reporters (1.5%) were included in the analysis.

### 2.4. Food Grouping

Food items were grouped into 17 categories (Appendix A), on the basis of the GloboDiet^®^’s 19 categories. Briefly, the “Legumes” and “Vegetables” categories from GloboDiet^®^ were grouped, as well as the “Cereals” and “Starchy” categories. Items from the “Meat” category were split into newly defined “White meat” and “Red and processed meat” categories. Food items labels, i.e., “Meat substitutes” and “Milk substitutes”, were removed from their GloboDiet^®^ categories (“Meat” and “Milk”, respectively) and were added to the “Others” category. Finally, non-caloric drinks were removed from the “Non-alcoholic beverages” category, with the exception of artificially sweetened soft drinks.

### 2.5. Identification of Dietary Patterns

The amounts of foods consumed (in g) were summed within each of the 17 categories and over each 24HDR. To account for differences in total energy intake between participants, the amounts of foods consumed were standardised by the total energy intake recorded in each 24HDR, and thus expressed in g/1000 kcal. We used a two-step procedure to identify dietary patterns. First, we used multiple factorial analysis (MFA) [23,24,25,26], an extension of principal component analysis tailored to handle multiple data tables. Thus, MFA was applied to the energy-standardised food consumption, considering the two 24HDR as two separate observations of the same individuals. This strategy allowed preserving the structure of each 24HDR rather than averaging food consumption over the two 24HDRs. In addition, as individuals with very low or very high energy intake showed extreme values of energy-standardised food consumption, individuals within the first and last percentiles of total energy intake in one of the two 24HDR (*n* = 82) were considered as supplementary individuals in the MFA (i.e., they did not influence the identification of principal components). Multiple criteria were used to define the number of principal components to retain (see Appendix A for summary). In a second step, the seven first principal components were used as inputs to hierarchical clustering using the Ward’s criterion [27]. While the number of clusters to retain is a debated topic [28], we used the decrease in within-inertia from *n* to *n*+1 cluster, and the interpretability of the partition, to choose four clusters (Appendix A). The defined partition was further consolidated using a k-means algorithm.

### 2.6. Sociodemographic, Lifestyle and Anthropometric Variables

Participants answered a questionnaire on the day of the first 24HDR (available in French, German or Italian [29]), which allowed considering the potential following determinants of dietary patterns: nationality (Swiss nationality only, Swiss binationality, and other nationality), education (originally in 19 categories; grouped into primary, secondary, and tertiary education), marital status (single, married or in registered partnership, divorced or terminated registered partnership, and other including widows), gross household income (<6000, between 6 to 13,000, and above 13,000 Swiss Francs/month), smoking status (never, former, and current), overall health status (originally in 5 levels; grouped into very bad to medium, and good to very good), and currently following a weight-loss diet (yes, and no) were self-declared. The age of the participants was calculated using the self-declared birth date and controlled using the age provided by the Federal Statistical Office (18–29, 30–44, 45–59, and 60–75 years old). The language region was determined by the canton of residence of the participants (German-speaking: Aargau, Basel-Land, Basel-Stadt, Bern, Lucerne, St. Gallen, and Zurich; French-speaking: Geneva, Jura, Neuchatel, and Vaud; Italian-speaking: Ticino). Participants’ physical activity was assessed using the short-form International Physical Activity Questionnaire (IPAQ) [30,31] and categorised into three physical activity levels following IPAQ classifications: low, moderate and high [32]. Finally, body weight, height and waist circumferences were measured following international standard protocols [33] as described elsewhere [14]. Body mass index (BMI) was calculated on the basis of these measures, except for pregnant (*n* = 14) and lactating (*n* = 13) women (self-reported values of weight before pregnancy) or when measurements were impossible (*n* = 7).

### 2.7. Determinants of Dietary Patterns

To increase the number of individuals available for regression, we performed multivariate imputation by chained equations (*m* = 25) [34]. Multinomial logistic regression was used to predict the probability of belonging to one of the dietary patterns using the following variables: sex, age, BMI, language region, nationality, education, marital status, gross household income, physical activity, smoking, overall health status and currently on a diet.

### 2.8. Weighting

To correct for the sampling design and non-response, survey results were weighted for age, sex, marital status, major area of Switzerland, nationality and household size. In addition, because 24HDRs were unevenly collected over the year and over each week [15], analyses of dietary patterns, food consumption and macronutrient intakes were also weighted for seasonality (in 4 seasons, according to the mean date of the two 24HDRs) and days of the week (1. two 24HDRs between Monday and Thursday, 2. two 24HDRs between Friday and Sunday, 3. one 24HDR between Monday and Thursday and the other between Friday and Sunday). Further details of the weighting strategy are available online on a public data repository [29].

### 2.9. Software and Packages

All descriptive and statistical analyses were conducted with R (version 3.3.2 for Mac). MFA and hierarchical clustering were performed with the FactoMineR package [35]. The multinomial regression used the nnet package [36]. Visualisation of the predicted probability of belonging to a dietary pattern was plotted with the effects package [37]. The mice package was used to perform multivariate imputation by chained equations [34].

### 2.10. Reporting

We used the guidelines of the STROBE checklists for cross-sectional study [38] and nutritional epidemiology [39] to report the present findings.

## 3. Results

### 3.1. Population Characteristics

Table 1 summarises the sociodemographic, lifestyle and athropometric characteristics of the study population and indicates the impact of statistical weighting for sex, age, marital status, major area, household size and nationality on the overall population structure. The study included 2057 participants, who represented a total population of 4,627,878 individuals after weighting. The majority of the participants were Swiss citizens (61.4% with Swiss citizenship and 13.8% Swiss binationals), had normal BMI (54.1%) and self-reported a good-to-very good health status (87.1%).

### 3.2. Dietary Patterns: Food Consumption

Using MFA and hierarchical clustering, we determined four dietary patterns on the basis of centre-reduced energy-standardised consumptions of the 17 food categories (Appendix A). Figure 1 summarises the four dietary profiles relative to the overall population (z-scores) and Figure 2 indicates the absolute energy-standardised consumption of each food category (in g/1000 kcal). It can be seen that 36.2% of the study population (*n* = 744) followed a “Swiss traditional” pattern characterised by minimal variation to the average of the population for all food categories (z-score ≈ 0), except for chocolate and milk and dairy consumptions (Figure 1 and Figure 2). We found two “Western” patterns representing 18.6% and 21.6% of the population, both characterised by a high intake of red and processed meat (Figure 1 and Figure 2). The first “Western” pattern, however, was characterised by a high intake of non-alcoholic beverages (274.4 g/1000 kcal; mostly driven by soft drinks consumption), whereas the second “Western” pattern was characterised by a high intake of alcoholic drinks (168.5 g/1000 kcal) and cereals and starchy food (157.4 g/1000 kcal; Figure 1 and Figure 2). 23.6% of the population (*n* = 486) were categorized as a “Prudent” dietary pattern, characterised by a high intake of fruits, vegetables, white meat and fish (Figure 1 and Figure 2).

### 3.3. Dietary Patterns: Macronutrient Intake

Figure 3 illustrates the overall energy intake, as well as the contributions to energy intake of the macronutrients, fibres and alcohol. As expected from the food consumption profiles, individuals following the “Western 1” pattern had the highest calorie intake from carbohydrates (1019.1 kcal vs. 940.5 kcal in the overall study population). Individuals following the “Western 2” pattern showed the highest calorie intake from alcohol (171.8 kcal vs. 90.9 kcal in the overall study population). Finally, individuals following a “Prudent” dietary pattern showed the lowest total energy intake compared to the three other dietary patterns (1944 kcal vs. 2226 kcal in the overall study population).

### 3.4. Sociodemographic and Lifestyle Determinants of Dietary Patterns

Appendix A describes the sociodemographic, lifestyle and anthropometric characteristics of the population by dietary pattern followed. A multinomial logistic regression on all potential determinants included in the study indicated that sex, age, BMI, language region, nationality, marital status, gross household income, physical activity, smoking status and being on a weight-loss diet are significant determinants of diets (Table 2). Notably, females showed a reduced predicted probability (prob) to follow one of the “Western” patterns compared to men (combined probability of “Western 1” and “Western 2” patterns: prob = 0.31 vs. prob = 0.50), older adults (> 60 years old) showed an increased probability to follow a “Prudent” pattern compared to younger ones (prob = 0.34 vs. prob = 0.17 to 0.27 in younger age groups), and Swiss citizens showed an increased probability to follow a Swiss traditional patterns than non-Swiss citizens (prob = 0.40 vs. prob = 0.25) (Figure 4). The sociodemographic variables included in the multinomial logistic regression, however, showed a limited ability to predict dietary patterns (Appendix A), indicating that a large part of the variation in dietary patterns remains unexplained.

## 4. Discussion

Applying MFA and hierarchical clustering on the data from the first national nutrition survey *menuCH*, we found 4 discrete dietary patterns in a sample of the population of Switzerland. Using multinomial logistic regression, we identified ten sociodemographic and lifestyle determinants of dietary patterns, indicating that the range of factors influencing dietary choices is broader than the previously reported sex and regional differences in food consumption in Switzerland [15].

Two of the dietary patterns found in Switzerland are consistently reported in industrialized countries as “Western” patterns [40,41,42]. This pattern generally features high consumptions of red and processed meat, refined sugars, starchy foods and high-fat products [6]. Interestingly, in our sample from Switzerland, we found two distinct patterns which only partially matched this definition. The “Western 1” pattern was characterised by a high intake of red and processed meat and refined sugars (in non-alcoholic beverages and cakes), which translated into the highest absolute energy intake from carbohydrates (1019.1 kcal) of all clusters. The “Western 2” pattern was also characterised by a high consumption of red and processed meat but also showed a high intake of cereals and starchy products, as well as sauces and seasonings. A discrepancy was visible between the two patterns in the intake of beverages (non-alcoholic beverages—mostly caloric soft drinks—in “Western 1” and alcoholic beverages in “Western 2”). Despite these differences, the two “Western” patterns largely shared the classic sociodemographic determinants of a Western diet [6,8] (males, young age, high BMI, and current smokers). Surprisingly, however, we observed a higher predicted probability for Italian-speaking individuals to follow the “Western 2” pattern, although no significant differences in alcohol or cereal and starchy products were reported between the Italian-speaking and other regions of Switzerland [15]. Additionally, an alcohol/meat dietary pattern has classically been reported in French nutrition studies [43,44,45], and might have been more expected in the French-speaking region of Switzerland. Interestingly, in our study, education was not a significant determinant of Western patterns, contradicting the common assumption that a Western diet is strongly associated with a low socio-economic status [46,47,48].

Interestingly, one of the identified patterns (“Swiss traditional”) was characterised with high consumption of milk and dairy products, as well as high chocolate intake, which were previously identified as core features of Swiss diets [49]. The predicted probability to follow a Swiss traditional diet was higher among Swiss nationals compared to that for non-Swiss citizens, and in the German-speaking regions compared to the French- or Italian-speaking regions. This confirms the assumption that such a diet is a feature of the Swiss culture, but indicates that it is more characteristic of German-speaking regions. Similar results were observed in another Swiss study, where higher milk, yoghurt and chocolate consumptions were observed in the German-speaking regions compared to other regions of Switzerland [15]. In addition, the consumption of milk and dairy products was twice as low in all the other three dietary patterns as in the “Swiss traditional” pattern, indicating that the majority of the population has a non-traditional attitude towards the consumption of milk and dairy products. This likely explains that adherence to the Swiss recommendations related to the consumption of milk and dairy products (three portions per day) remains low in the overall population (21.7% [15]).

Overall, we found that the language region is a main determinant of dietary patterns, which confirms the major differences in consumption of numerous individual foods across language regions of Switzerland reported previously [15]. It is likely that dietary patterns seen in Swiss language regions are differentially influenced by the dietary habits of the larger neighboring countries (France, Italy, Germany and Austria). Indeed, dietary patterns of France, Italy and Germany share numerous dietary characteristics with their corresponding language regions of Switzerland [50]. For example, the Italian dietary pattern is characterised by a high intake of cereals and starchy products compared to the EPIC mean [50], which is in accordance with the high probability for individuals of the Swiss Italian-speaking region to follow the “Western 2” pattern in our study. In the same study, the German diet was characterised by a low vegetable and fruit intake (below the EPIC mean), which echoes the smaller probability for residents of a German-speaking region to follow a “Prudent” pattern compared to residents of other language regions. Our dietary pattern approach based on relatively broad food categories, however, did not allow detecting more qualitative differences classically reported between a German-like and a Mediterranean-like diet, such as the use of animal vs. plant-based fats [51], or the type of beverages consumed [5]. In a more holistic dietary assessment, dietary patterns of the *menuCH* participants were further described by our group using the a priori-defined Alternate Healthy Eating Index and Mediterranean Diet Score [52]. Altogether, our findings further highlight the differential influence of the dietary cultures of neighboring countries on the language regions of Switzerland, and are reminiscent of other multilingual countries such as Belgium or Canada [53,54].

The consumption of different dietary patterns is associated with differential nutrient intakes between and within European countries, as emphasised by the EPIC study [55]. In the *menuCH* study, we observed large variation in energy intake and intake of macronutrients across dietary patterns. The “Western 1” pattern showed the highest energy intake of all four dietary patterns and obesity was associated with this dietary pattern in the multinomial logistic regression. Further information on micronutrient intake, however, was partially missing in the current Swiss Food Composition Database but may be useful to further investigate the nutritional quality of each pattern. Notably, this will be important to correlate nutrient intake with geographic patterns of mortality in Switzerland [4].

Finally, the multinomial logistic regression model using sociodemographic variables showed a limited ability to predict dietary patterns in our sample from Switzerland (Appendix A). This indicates that other factors (either sociodemographic or not) determine dietary choices in Switzerland, such as constraints (time, costs or food availability, etc.), attitudes or knowledge [56]. Interestingly, these factors (notably, taste, price and time) were also identified as barriers to healthy eating in Switzerland [11]. In the same study, these barriers to healthy eating were shown to evolve dynamically across all sociodemographic groups. Accordingly, changes in diets were also shown to depend on sociodemographic determinants [57] and to evolve over time [58]. Altogether, these findings suggest that the sociodemographic determinants of diets found in this study may be time-dependent. 

A major strength of our study is the quality of the analysed data set. First, it allows for generalizability of the results to the national level. Indeed, the *menuCH* study population is based on a national stratified random population-based sample. Moreover, statistical weighting allowed correcting for changes in population structure due to non-response and for the uneven distribution of 24HDR across weekdays and seasons. A second strength is the use of state-of-the-art dietary assessment tools, limiting the probability of reporting bias: multiple pass 24HDRs using the reference software in European food consumption surveys, GloboDiet^®^ [19], were conducted by trained dietitians, and were followed by data quality control according to a tested scheme [14]. The use of two 24HDRs, if well suited to providing estimates of food consumption at the population level, may not accurately reflect habitual intakes at the individual level: therefore, participants to the *menuCH* study may have been wrongly assigned to a dietary pattern. Non-differential misclassification, however, may rather lead to an underestimation of the association between dietary patterns and the sociodemographic and lifestyle determinants. In addition, a challenge we faced was the absence of gold standard data-driven method to derive dietary patterns from two 24HDRs. Like a few others [59,60,61], we used a two-step strategy in the identification of dietary patterns, following up MFA with hierarchical clustering. This strategy allows first for removal of the non-interpretable variation (“noise”) in diets and second for the identification of discrete dietary patterns on the remaining interpretable variation [59]. Supplementary dietary patterns might have been eliminated during the pre-processing step with MFA, but these patterns would not be clearly supported by the data and would be difficult to interpret. We acknowledge that both principal components and clustering methods are partially based on arbitrary analytic decisions (number of dimensions or clusters to retain, for example [28]) and may show limited stability [62,63] or reproducibility [64,65,66,67].

## 5. Conclusions

In conclusion, our study identified sociodemographic and lifestyle determinants of dietary patterns in Switzerland. A unique finding of our study is that the language region is a major determinant of diets. Strikingly, the distribution of dietary patterns between language regions appears to reflect the cultural influence of the respective neighbouring countries (Germany, Italy, and France). Together, our findings have the potential to foster and help design public health interventions targeted to population subgroups, including language region-specific campaigns.

## Figures and Tables

**Figure 1 nutrients-11-00062-f001:**
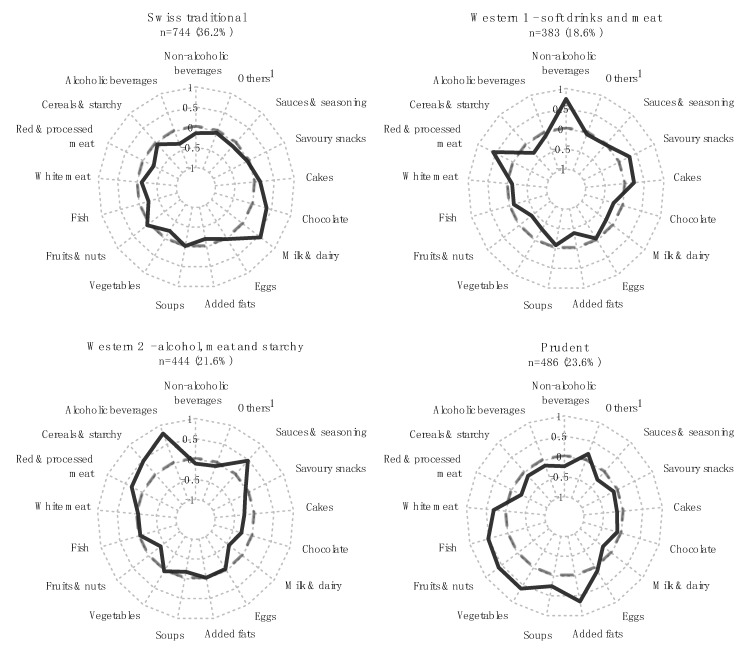
Food consumption profiles in the four dietary patterns relative to the overall population. Energy-standardised consumptions of the 17 food categories were centred and reduced for the overall study population (z-standardisation). Each axis of the radar plots indicates the mean of the centred-reduced energy-standardised consumptions of one food category within one dietary pattern, i.e., how the consumption in a dietary pattern deviates from the consumption in the overall population. A positive and a negative value indicate consumptions above and below the mean of the overall population, respectively. ^1^ Others include meat substitutes, milk substitutes and meal replacements (all categories are described in Appendix A).

**Figure 2 nutrients-11-00062-f002:**
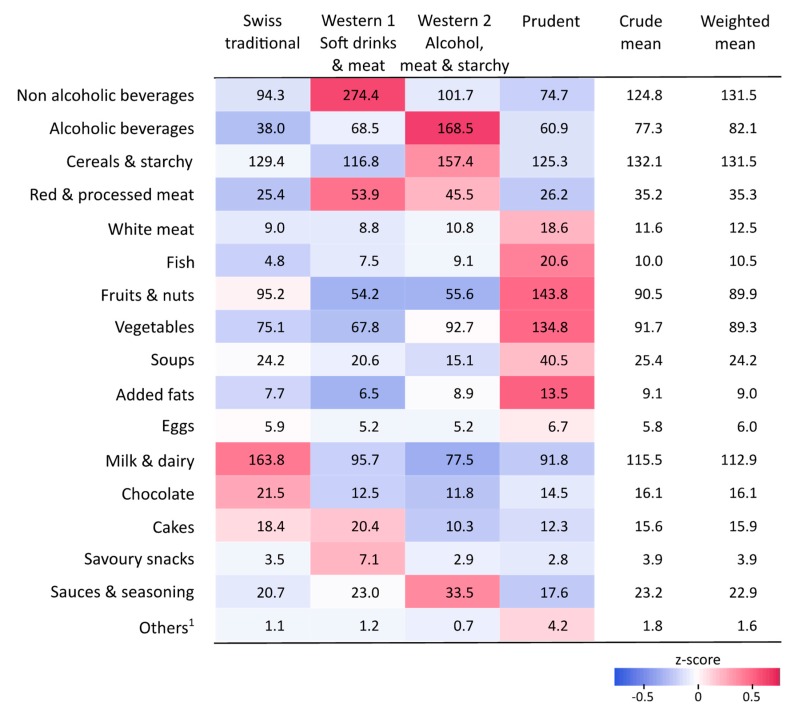
Amounts of food consumed in the four dietary patterns and the overall population and energy-standardised consumptions of the 17 food categories (in g/1000 kcal) in the four dietary patterns and the overall study population. The weighted mean provides estimates of the mean consumption corrected for sex, age, marital status, major area, household size, nationality and the uneven distribution of 24-h dietary recalls over seasons and weekdays. Colors indicate the mean of the energy- and z-standardised consumption of one food category within one dietary pattern. ^1^ Others include meat substitutes, milk substitutes and meal replacements (all categories are described in Appendix A).

**Figure 3 nutrients-11-00062-f003:**
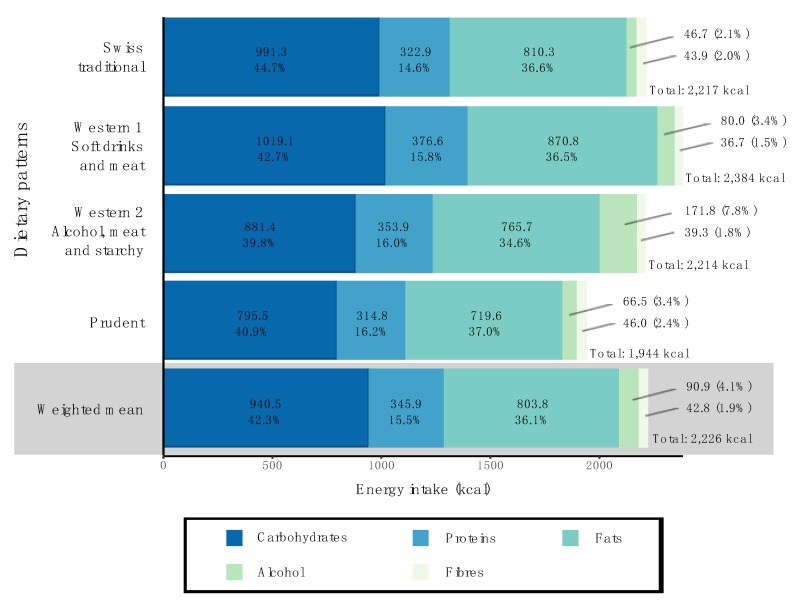
Macronutrient contributions to total energy intake in the four dietary patterns and the overall study population. Figures indicate the mean energy (in kcal) brought by macronutrients, alcohol and fibres and their contributions to the total energy intake. The weighted mean provides estimates of the mean energy in the study population after correction for sex, age, marital status, major area, household size, nationality and the uneven distribution of 24-h dietary recalls over seasons and weekdays.

**Figure 4 nutrients-11-00062-f004:**
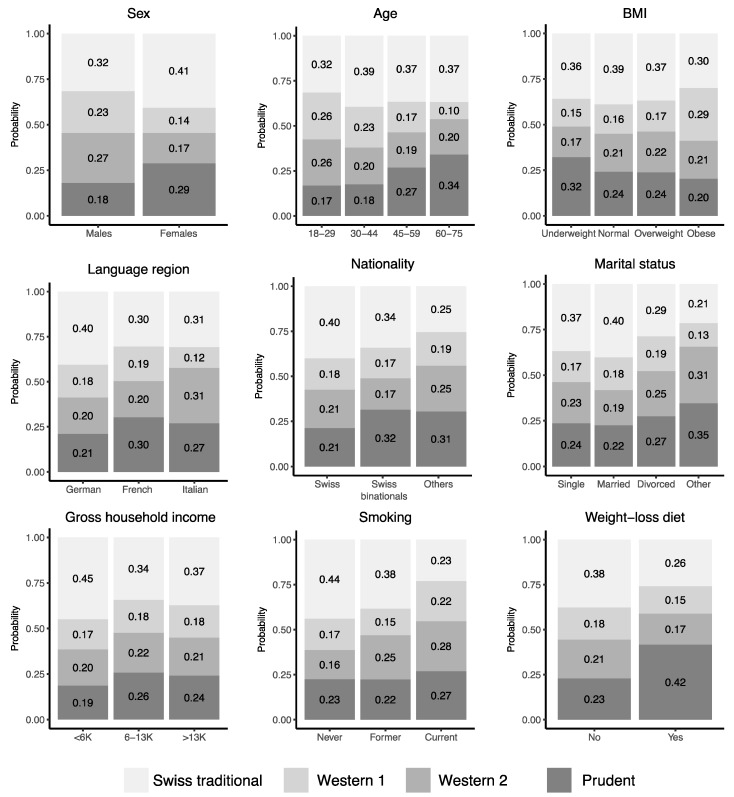
Effect plots of the main determinants of dietary patterns in Switzerland. Stacked bar plots indicate the probability for individuals to follow a dietary pattern as predicted by the multinomial logistic regression model (sex, age groups, BMI, language region, nationality, education, marital status, gross household income, self-reported physical activity, smoking, self-reported health, and weight-loss diet; all results are described in Table 2). Abbreviation: BMI, Body Mass Index.

**Table 1 nutrients-11-00062-t001:** Characteristics of the study participants.

	Crude	Weighted ^1^
**Participants with 2 complete 24HDRs; *n***	2057	
**Sum of weights for weighted analysis; *n***	-	4,627,878
**Sex**		
Males	45.4%	49.8%
Females	54.6%	50.2%
**Age groups ^2^**		
18–29 years old	19.4%	18.8%
30–44 years old	25.9%	29.9%
45–59 years old	30.4%	29.8%
60–75 years old	24.3%	21.6%
**BMI categories ^3^**		
Underweight (BMI < 18.5 kg/m^2^)	2.5%	2.4%
Normal weight (18.5 ≤ BMI < 25 kg/m^2^)	54.2%	54.1%
Overweight (25 ≤ BMI < 30 kg/m^2^)	30.6%	30.6%
Obese (BMI ≥ 30 kg/m^2^)	12.7%	12.9%
**Language region ^4^**		
German-speaking	65.2%	69.2%
French-speaking	24.4%	25.2%
Italian-speaking	10.4%	5.6%
**Nationality**		
Swiss	72.5%	61.4%
Swiss binationals	14.4%	13.8%
Other	13.0%	24.8%
**Education, highest degree**		
Primary school or no degree	4.3%	4.7%
Secondary	47.1%	42.6%
Tertiary	48.5%	52.6%
**Marital status**		
Single	30.8%	31.1%
Married or in registered partnership	54.7%	52.2%
Divorced or terminated partnership	10.8%	12.1%
Other	3.5%	4.4%
**Gross household income (CHF/month)**		
<6000	16.8%	17.7%
6000 to 13,000	40.9%	39.8%
>13,000	13.9%	14.9%
Imputed	28.4%	27.6%
**Self-reported physical activity**		
Low	12.2%	12.9%
Moderate	22.1%	22.7%
High	40.2%	40.3%
Imputed	25.5%	24.2%
**Smoking**		
Never smoker	44.4%	42.9%
Former smoker	33.4%	33.6%
Current smoker	21.9%	23.3%
**Self-reported health**		
Very bad to medium	13.2%	12.7%
Good to very good	86.6%	87.1%
**Currently on a weight-loss diet**		
No	94.3%	94.4%
Yes	5.5%	5.4%

^1^ Percentages are weighted for sex, age, marital status, major area, household size, and nationality. ^2^ Age is the self-reported age on the day the dietary and physical activity behavior questionnaire was filled. ^3^ BMI was obtained from measured height and weight. Self-reported weight or height was used when measurements were impossible. For lactating and pregnant women, self-reported weight before pregnancy was used to calculate BMI. ^4^ German-speaking regions: the cantons of Aargau, Basel-Land, Basel-Stadt, Bern, Lucerne, St. Gallen, and Zurich; French-speaking regions: Geneva, Jura, Neuchatel, and Vaud; and Italian-speaking region: Ticino. The numbers of imputed values are not shown for variables with less than 0.2% of missing values (0 to 4). Abbreviations: 24HDR, 24-h dietary recall; BMI, Body Mass Index; CHF, Swiss Francs.

**Table 2 nutrients-11-00062-t002:** Association between dietary patterns and sociodemographic and lifestyle factors.

	“Western 1”—Soft Drinks and Meat		“Western 2”—Alcohol, Meat and Starchy		“Prudent”	
	OR	95% CI	OR	95% CI	OR	95% CI
**Sex**						
Males (reference)	1		1		1	
Females	**0.47**	**[0.35–0.61]**	**0.47**	**[0.36–0.61]**	1.24	[0.95–1.61]
**Age groups ^1^**						
18–29 years old	1.43	[0.93–2.18]	**1.57**	**[1.03–2.39]**	1.20	[0.78–1.84]
30–44 years old (reference)	1		1		1	
45–59 years old	0.80	[0.57–1.13]	1.03	[0.73–1.46]	**1.65**	**[1.19–2.29]**
60–76 years old	**0.45**	**[0.29–0.71]**	1.04	[0.70–1.55]	**2.08**	**[1.43–3.03]**
**BMI categories ^2^**						
Underweight (BMI < 18.5 kg/m^2^)	1.02	[0.41–2.53]	0.88	[0.36–2.11]	1.45	[0.70–2.98]
Normal (18.5 ≤ BMI < 25 kg/m^2^)—(reference)	1		1		1	
Overweight (25 ≤ BMI < 30 kg/m^2^)	1.11	[0.82–1.51]	1.13	[0.85–1.52]	1.04	[0.78–1.39]
Obese (BMI ≥ 30 kg/m^2^)	**2.32**	**[1.54–3.50]**	1.30	[0.85–2.00]	1.09	[0.72–1.67]
**Language region ^3^**						
German-speaking (reference)	1		1		1	
French-speaking	**1.40**	**[1.03–1.91]**	1.33	[0.98–1.80]	**1.92**	**[1.45–2.53]**
Italian-speaking	0.83	[0.43–1.61]	**2.00**	**[1.19–3.38]**	1.68	[0.98–2.90]
**Nationality**						
Swiss (reference)	1		1		1	
Swiss binationals	1.13	[0.77–1.68]	0.96	[0.64–1.43]	**1.73**	**[1.22–2.45]**
Others	**1.67**	**[1.20–2.33]**	**1.88**	**[1.36–2.59]**	**2.25**	**[1.64–3.08]**
**Education, highest degree**						
Primary school or no degree	0.90	[0.46–1.77]	0.85	[0.45–1.60]	1.15	[0.62–2.12]
Secondary (reference)	1		1		1	
Tertiary	0.77	[0.58–1.02]	0.79	[0.61–1.04]	1.25	[0.96–1.63]
**Marital status**						
Single (reference)	1		1		1	
Married or in registered partnership	0.97	[0.67–1.39]	0.78	[0.55–1.11]	0.87	[0.62–1.22]
Divorced or terminated partnership	1.44	[0.85–2.45]	1.41	[0.86–2.33]	1.49	[0.92–2.40]
Other	1.30	[0.57–2.98]	**2.36**	**[1.20–4.64]**	**2.51**	**[1.31–4.80]**
**Gross household income (CHF/month)**						
<6000	**0.70**	**[0.49–1.00]**	**0.69**	**[0.49–0.97]**	**0.55**	**[0.39–0.78]**
6000 to 13,000 (reference)	1		1		1	
>13,000	0.91	[0.63–1.31]	0.88	[0.61–1.26]	0.86	[0.62–1.21]
**Self-reported physical activity**						
Low (reference)	1		1		1	
Moderate	0.72	[0.48–1.09]	**1.73**	**[1.12–2.67]**	1.15	[0.77–1.71]
High	0.88	[0.61–1.27]	**1.51**	**[1.01–2.27]**	1.00	[0.69–1.45]
**Smoking**						
Never smoker (reference)	1		1		1	
Former smoker	0.97	[0.71–1.31]	**1.75**	**[1.31–2.34]**	1.14	[0.87–1.49]
Current smoker	**2.42**	**[1.72–3.40]**	**3.30**	**[2.34–4.64]**	**2.28**	**[1.62–3.20]**
**Self-reported health**						
Very bad to medium	1.06	[0.70–1.61]	1.38	[0.94–2.03]	1.05	[0.71–1.55]
Good to very good (reference)	1		1		1	
**Currently on a diet**						
No (reference)	1		1		1	
Yes	1.25	[0.67–2.33]	1.17	[0.62–2.20]	**2.65**	**[1.56–4.51]**

Results of the multinomial regressions were adjusted for all variables presented in this table, and weighted for sex, age, marital status, major area, household size, and nationality. The “Swiss Traditional” group was used as a reference. ^1^ Age groups are based on the self-reported age on the day the dietary and physical activity behavior questionnaire was filled. ^2^ BMI was obtained from measured height and weight. Self-reported weight or height was used when measurements were impossible. For lactating and pregnant women, self-reported weight before pregnancy was used to calculate BMI. ^3^ German-speaking regions: Aargau, Basel-Land, Basel-Stadt, Bern, Lucerne, St. Gallen, and Zurich; French-speaking regions: Geneva, Jura, Neuchatel, and Vaud; and Italian-speaking region: Ticino. ORs in bold are associated with a *p*-value of <0.05. Abbreviations: 24HDR, 24-h dietary recall; BMI, Body Mass Index; CHF, Swiss Francs; CI, Confidence Interval; OR, Odds Ratio.

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
