# Peer review of "Dietary Patterns and Their Sociodemographic and Lifestyle Determinants in Switzerland: Results from the National Nutrition Survey menuCH"

_nutrients, 2018, doi:10.3390/nu11010062_

Round 1
Reviewer 1 Report
The manuscript was well written, the authors presented details in the methods and results of pattern identification. The Tables and Figures are clear. Only two-day 24HDR cannot represent the long-term pattern of food consumption, and might probably bias the results. Anyway, the data are worthy of the publication.
I only had a few comments:
1. The lack of representativeness of two 24HDR is better to be discussed as one of the limitation.
2. The authors did not provide the information of which season the 24HDR were recorded. If two 24HDR of all participants were not well distributed in different seasons, the data cannot represent the food consumption of the whole year.
3. The authors create an “others” category to include “milk and yogurt substitutes” and “meat substitutes”, I doubted if it is a good strategy to do so. If milk substitutes and meat substitutes were high-frequently consumed, it is better to be separated. Also the authors may try to use more categories instead of only 17 categories.
4. It is more common omit the individuals with very low and very high energy intake instead of keeping them in the data set, and it is also more common to define the extreme energy intake according to the value relative to the recommended reference dose of certain population.
5. Physical activity can be measured by Metabolic Equivalent of Task (MET).
6. Why the authors used gross household income instead of income at individual level? In some cases household income were high and the family numbers are also high, the family cannot spend too much money to buy the food.
Author Response
Answers to Reviewer 1 can be found in the attached PDF document (we provided tables of sensitivity analyses that would not be readable if pasted here!).

Reviewer 2 Report
This article is very well written and discusses the results of dietary pattern analysis conducted on data from a representative diet survey in Switzerland. The results are potentially useful for public health nutritionists in Switzerland, and those interested in the effects of sociodemographic and lifestyle factors on dietary patterns.
I have only a few small comments because the authors have been very thorough in the preparation of their manuscript.
Abstract: include a very brief description of what a "Swiss traditional" diet is.
Lines 73 to 75: Were the demographics of people who agreed to participate different to those who did not agree to participate, or those who were not able to be contacted? Please comment briefly in the text.
Line 132: Please elaborate on the "other" category, for example is this only for widows/widowers, or do participants in common law/de facto arrangements also belong here?
General: While in Swiss cantons such as Basel and Zurich, the predominant language spoken is German, a significant minority of inhabitants speak either French or Italian due to migration from other Cantons, or from neighbouring countries. It would be interesting to see whether participants' main language spoken (which could perhaps be part of the dataset as the language in which the questionnaire was given) is a stronger predictor of dietary pattern than regional language.
Author Response
Please read our answers in the attached PDF document (we provided tables that could not be paster here).
